# Development of Single Nucleotide Polymorphism and Phylogenetic Analysis of *Rhododendron* Species in Zhejiang Province, China, Using ddRAD-Seq Technology

**DOI:** 10.3390/plants14101548

**Published:** 2025-05-21

**Authors:** Hong Zhu, Dongbin Li, Chunlei Yue, Hepeng Li

**Affiliations:** 1Research Centre for Zhejiang Wetland, Zhejiang Academy of Foresty, Hangzhou 310023, China; 84244@163.com (H.Z.); chunlei@chinaacc.com (C.Y.); 2Ningbo Forestry Development Center, Ningbo 315440, China; comeonldb@163.com

**Keywords:** ddRAD-seq, *Rhododendron* L., single nucleotide polymorphism, SNP, phylogenomic

## Abstract

The genus *Rhododendron* presents significant challenges for systematic classification due to extensive hybridization and adaptive radiation. Here, we employed double-digest restriction site-associated DNA sequencing (ddRAD-seq) to resolve phylogenetic relationships among nine ecologically significant *Rhododendron* species (34 accessions) endemic to Zhejiang Province, China, a biodiversity hotspot for this genus. Using *R. simsii* as the reference genome, we generated 39.40 Gb of high-quality sequencing data with a Q30 score of 96.65% and a GC content of 39.63%, achieving an average alignment rate of 92.79%. Through stringent filtering (QD ≥ 2, MQ ≥ 40), we identified 14,048,702 genome-wide single nucleotide polymorphism (SNP), predominantly characterized by the mutation types T:A>C:G and C:G>T:A. The widespread *R. simsii* and *R. simsii* var. *putuoense* exhibited significant genetic diversity, whereas the low-altitude widespread *R. molle* and the endemic *R. simiarum* exhibited lower genetic diversity. Moderate genetic differentiation (*F*_st_ = 0.097) was observed between *R. simsii* and *R. simsii* var. *putuoense*, while substantial genetic differentiation was detected among the other *Rhododendron* species. Principal component analysis (PCA), combined with phylogenomic reconstruction, demonstrated that the *Rhododendron* genus can be stratified into six well-supported genetic clades. Furthermore, this study provides the first genomic validation of the sibling relationship between *R. simsii* and its variety, *R. simsii* var. *putuoense*, and clarifies the systematic position of *R. huadingense*, suggesting that it should be classified as a new subgenus. This study establishes ddRAD-seq as a cost-effective tool, providing both a theoretical framework for SNP-based phylogenetics and critical insights for conserving China’s azalea biodiversity.

## 1. Introduction

*Rhododendron* L. (Ericaceae) comprises over 1000 species globally and is widely distributed across the temperate regions of Europe, Asia, and North America. China possesses the richest resources of *Rhododendron* species and is also one of the primary centers of modern distribution and differentiation of the genus worldwide [1]. The country is home to approximately 571 wild species, with more than 74% endemic to China [2,3]. As a vital member of subtropical forest ecosystems, *Rhododendron* plants play a significant role in maintaining the stability of the structure and function of mountainous ecosystems [4]. Additionally, this genus of plants holds considerable ornamental value [5], medicinal applications, and development potential [6].

Accurate construction of phylogenetic relationship is crucial for understanding species evolution. Due to habitat diversity and reproductive modes influenced by rapid adaptive radiation and extensive natural hybridization among species, the genus *Rhododendron* is regarded one of the most challenging plant groups. Hybridization and genome duplication have significantly shaped the diversification of *Rhododendron*, driving adaptive evolution through enhanced genetic variation and novel genomic architectures [7,8]. Previous studied have employed various molecular marker techniques to investigate the phylogeny and diversity of *Rhododendron* species, including DNA barcoding [9,10], simple sequence repeats (EST-SSRs) [11,12], plastid genomes [13,14], and restriction-site-associated DNA sequencing (RAD-seq) [15]. Although research on the genus *Rhododendron* has been conducted for many years, the aforementioned markers exhibit several limitations such as low throughput, time consumption, inaccuracy, and poor cost-effectiveness, which have led to numerous unresolved issues and restricted the scientific classification and accurate identification of *Rhododendron* species. In recent years, the rapid advancement of genome sequencing technology, molecular marker research has presented new opportunities. Double-digest restriction-site-associated DNA sequencing (ddRAD-seq) is a reduced-representation genome sequencing (RRGS) technique developed based on next-generation sequencing (NGS), capable of developing a large number of single nucleotide polymorphism (SNP) markers at a lower cost, independent of the presence of a reference genome or chromosomal ploidy [16]. Because SNP are distributed throughout the genome, they offer greater coverage and accuracy, thus offering higher resolution and reliability compared to traditional molecular markers.

Currently, a total of 19 wild species of the genus *Rhododendron* have been recorded in Zhejiang Province in China. Although this number is relatively small, these species encompass four subgenera: subgen. *Tsutsusi* (Sweet) Pojarkova, subgen. *Pentanthera* (G. Don) Pojarkova, subgen. *Azaleastrum* Planchon ex K. Koch, and subgen. *Hymenanthes* (Blume) K. Koch. Furthermore, ongoing discoveries of new local species, such as *Rhododendron huadingense* B. Y. Ding & Y. Y. Fang [17,18] and *R. simsii* var. *putuoense* G. Y. Li & Z. H. Chen [19], have introduced new species and varieties published after the release of the Flora of China (Volume 57) in 1994. However, the descriptions of these two species was based solely on morphological descriptions, lacking robust molecular evidence. This limitation has led to controversies regarding the systematic positions and taxonomic definitions of some species.

This study employs ddRAD-seq technology to perform reduced-representation genome sequencing of representative *Rhododendron* plants from Zhejiang, with the aim of identifying high-quality SNP markers at the genomic level. The objectives are as follows: (1) to evaluate the genetic diversity and differentiation among these species and (2) to reconstruct robust phylogenetic relationships, thereby providing evidence and solutions for the challenges associated with classification *Rhododendron* species in Zhejiang.

## 2. Materials and Methods

### 2.1. Plant Materials

A total of 34 samples representing nine species and one variety of *Rhododendron* were used for ddRAD-seq analysis. The species included *R. simsii* Planch., *R. farrerae* Sweet, *R. huadingense*, *R. molle* (Blume) G. Don, *R. ovatum* (Lindl.) Planch. ex Maxim., *R. championiae* Hook., *R. latoucheae* Franch., *R. simiarum* Hance, *R. fortunei* Lindl., and *R. simsii* var. *putuoense*. Two to five individuals were sampled from each taxon, encompassing four subgenera and seven sections following the Flora of China (2005) (Table 1). Among these, *R. simsii* var. *putuoense*, a variety published in 2010, has not yet been assigned an infrageneric taxon system. Field sampling was conducted in September 2024. Except *R. simsii* var. *putuoense*, which was collected from Dinghai District, Zhoushan (122.11° E, 30.08° N), all other specimens were obtained from the Jinhua Pan’an *Rhododendron* Villa (120.54° E, 28.39° N). This repository is dedicated to the conservation and study of native *Rhododendron* species, particularly those indigenous to Zhejiang Province.

### 2.2. DNA Extraction and Library Preparation

For each individual, 0.5 g of fresh leaf tissue was collected, and total genomic DNA was extracted using the CTAB method, followed by quality assessment with 1.2% agarose gel electrophoresis. A combination of restriction endonucleases *Dpn*II and *Msp*I was employed for genomic digestion, selecting DNA insert fragments ranging from 300 to 500 bp for the library. The library was sequenced using paired-end (PE) sequencing with 2 × 500 bp on the Illumina NovaSeq platform, with library construction and sequencing carried out by Shanghai Personalbio Biotechnology Co., Ltd., Shanghai, China.

### 2.3. SNP Calling

After sequencing data were generated, the fastp (v. 0.20.0) [20] was employed to remove adapter sequences and low-quality reads from the raw data using a sliding window method based on the following criteria: a 5 bp sliding filter was applied, removing windows with a Q value < 20, and trimming any reads with a length < 50 bp or containing ≥5 N bases at either end. The filtered sequences (clean reads) were aligned to the *R. simsii* reference genome (ASM1428224v1) using the mem program of BWA software (v. 0.7.12-r1039), followed by SNP detection using GATK software (v. 3.4.46) [21]. To ensure the reliability of the identified SNP loci, further filtering of the SNP genotype results was conducted, adhering to the following criteria: Fisher test of strand bias (FS) ≤ 60, Haplotype Score ≤ 13.0, Mapping Quality (MQ) ≥ 40, Quality Depth (QD) ≥ 2, Read Pos Rank Sum ≥ −8.0, and MQ Rank Sum > −12.5.

### 2.4. Genetic Diversity and Genetic Differentiation

Genetic diversity and differentiation analyses among species were conducted using the population command in the *Stacks* package, with parameters including: the average number of individuals per SNP (Num Indv), observed heterozygosity (*H*_o_), expected heterozygosity (*H*_e_), nucleotide diversity (π), inbreeding coefficient (*F*_is_), and pairwise fixation index (*F*_st_).

### 2.5. PCA and Phylogenomic Tree Reconstruction

Principal Component Analysis (PCA) was performed using GCTA software (v. 1.94.1) [22] on the genome-wide SNP dataset (14,048,702 loci) after filtering for minor allele frequency (MAF ≥ 0.05). Only the first two principal components (PC1 and PC2) were retained for visualization, as they cumulatively explained 25.97% of the genetic variance (see Results)”, retaining only the first two axes were retained for presentation. A phylogenomic tree was constructed using the Maximum Likelihood (ML) in fastTree software (v. 2.1.3) [23] and the reliability of the clades was validated using bootstrap resampling (1000 repetitions).

## 3. Results

### 3.1. Sequencing Data Quality Control and Filtering

ddRAD sequencing was conducted on 34 samples of *Rhododendron* species, resulting in a total of 39.40 Gb of data, with an average data volume of 1.16 Gb per sample. The number of generated raw reads ranged from 6,254,358 to 18,559,976, with an average GC content of 39.63%. The average Q20 and Q30 scores were 98.81% and 96.65%, respectively, indicating high sequencing quality. After filtering, 38.20 Gb of high-quality data was obtained, with an average data volume of 1.12 Gb per sample. The number of clean reads generated ranged from 6,100,640 to 18,210,666. The clean reads were aligned to the *Rhododendron* reference genome, with the number of aligned reads ranging from 5,642,084 to 17,107,014 and an alignment rate of 83.70% to 96.91%; the average alignment rate was 92.79%. The sequencing depth varied between 1.65× and 4.93×, and the alignment results met the criteria for subsequent data analysis (Table 1).

### 3.2. SNP Distribution and Identification

After filtering the SNP loci data, a total of 14,048,702 high-quality SNP loci were identified. The number of SNP were counted using a 100 kb sliding window across the populations. Based on the distribution across the 13 chromosomes, the SNP loci showed a relatively even distributed on the chromosomes (Figure 1).

Genomic SNP mutations can be categorized into six types, with the largest proportion being C:G>T:A, followed by T:A>C:G; these two types together account for 64.42%. The remaining four types collectively account for 35.58% (Figure 2).

### 3.3. Genetic Diversity Analysis

A statistical analysis of the genetic diversity parameters for ten species of *Rhododendron* was conducted (Table 2). The results indicate that the number of individuals (Num Indv) ranged from 1.9170 (*R. simiarum*) to 4.8102 (*R. simsii* var. *putuoense*), with a mean of 3.1966. Observed heterozygosity (*H*_o_) ranged from 0.0239 (*R. molle*) to 0.0549 (*R. molle*), with a mean of 0.0382. Expected heterozygosity (*H*_e_) ranged from 0.0199 (*R. molle*) to 0.0611 (*R. simsii*), with a mean of 0.0359; Nucleotide diversity (π) ranged from 0.0241 (*R. molle*) to 0.0685 (*R. simsii*), with a mean of 0.0426. Inbreeding coefficient (*F*_is_) ranged from −0.0066 (*R. simiarum*) to 0.0466 (*R. simsii*), with a mean of 0.0193. Four species (*R. molle*, *R. huadingense*, *R. championae*, and *R. simiarum*) exhibited negative *F*_is_ values, while the others had positive values.

### 3.4. Genetic Differentiation Analysis

The *F*_st_ values for the ten species of *Rhododendron* range from 0.097 to 0.806, with an average of 0.641. Among these species, the greatest genetic differentiation is observed between *R. molle* and *R. huadingense*, while the smallest genetic differentiation occurs between *R. simsii* var. *putuoense* and *R. simsii* (*F*_st_ = 0.097). *R. molle* (*F*_st_ range: 0.681–0.806) and *R. huadingense* (*F*_st_ range: 0.732–0.757) show strong differentiation with the other species (Figure 3).

To further investigate the genetic differentiation among the nine species and one variety of the genus *Rhododendron*, a PCA was conducted. The contribution rates of the first principal component (PC1) and the second principal component (PC2) were 13.87% (eigenvalue = 4.52) and 12.10% (eigenvalue = 3.94), respectively, resulting in a cumulative contribution rate of 25.97%. Based on the positions and mutual distances of the species in the two-dimensional plot, they can be broadly categorized into six groups (Figure 4).

### 3.5. Phylogenetic Analysis

Phylogenetic trees based on 14,048,702 SNP loci were constructed using the ML method, revealing that the 34 samples clustered into six well-supported genetic clades. The first clade comprised *R. simsii* var. *putuoense*, *R. simsii*, and *R. farrerae*, which belong to subgen. *Tsutsusi*. The second clade included only *R. ovatum*, which is classified under subgen. *Azaleastrum*. The third clade contained solely *R. huadingense*. The fourth clade encompassed *R. championiae* and *R. latoucheae*, which are classified under subgen. *Azaleastrum*. The fifth clade consisted solely of *R. molle*, belonging to subgen. *Pentanthera* and the sixth calde comprised *R. simiarum* and *R. fortunei*. These clustering results are consistent with the findings from the PCA (Figure 5).

## 4. Discussion

### 4.1. Genetic Diversity and Genetic Differentiation of Rhododendron Species Produced in Zhejiang Province

Genetic diversity is the result of long-term evolution in biological populations and acts as a prerequisite for their survival, adaptation, and development. Higher genetic diversity and richer variations enhance the ability to adapt to environmental changes. This study demonstrates that at the species level, *R. simsii* and *R. simsii* var. *putuoense* exhibit high genetic diversity, whereas *R. molle* and *R. simiarum* show low genetic diversity. The *Rhododendron* species *R. simsii*, *R. simsii* var. *putuoense*, and *R. molle* have been described as widely distributed across Zhejiang Province [24]. However, *R. molle* is confined to low hills and mountains below 1000 m in elevation, while *R. simiarum* is classified as an endemic species, restricted to high-altitude regions. For widely distributed species, larger population sizes and the potential for gene flow among populations across different regions further promote increased genetic diversity. In contrast, endemic species with restricted distribution ranges and smaller population sizes are more susceptible to genetic drift, which reduces genetic variation. Additionally, their adaptation to stable environments with lower selective pressures may further limit the generation and maintenance of novel genetic diversity, resulting in overall lower genetic diversity compared to widespread species.

As a classical metric for quantifying population divergence, pairwise *F*_st_ provide critical insights into genetic differentiation. Following Wright’s categorization (*F*_st_ < 0.05: negligible; 0.05–0.15: moderate; 0.15–0.25: considerable; >0.25: substantial) [25], the moderate differentiation (*F*_st_ = 0.097) values for *R. simsii* var. *putuoense* and *R. simsii* likely reflect a combination of ancestral gene flow and incipient divergence. Phylogenomic evidence supports their recent divergence from a common ancestor (Figure 5), with ongoing but limited gene flow, potentially maintained through historical range overlaps or occasional hybridization. However, their current parapatric distributions—*R. simsii* in mountainous habitats versus *R. simsii* var. *putuoense* in coastal island ecosystems—create ecological barriers that promote adaptive differentiation. In contrast, strong genetic boundaries (*F*_st_ > 0.25) among other congeneric species imply reproductive isolation, potentially driven by long-term geographic separation, ecological specialization, or intrinsic postzygotic barriers.

### 4.2. Phylogenetic Relationships and Taxonomy of Rhododendron Species Produced in Zhejiang Province

Despite numerous phylogenetic analyses in the genus *Rhododendron*, persistent controversies remain regarding its infrageneric classification. Early molecular studies relying on limited genomic markers [9,10], and restricted taxon sampling failed to resolve deep phylogenetic relationships. Our phylogenomic analysis confirms the monophyly of subgen. *Tsutsusi*, encompassing sect. *Tsutsusi* and sect. *Brachycalyx* (represented by *R. farrerae*). Contrasting with classical taxonomy, we reveal the polyphyly of subgen. *Azaleastrum*: sect. *Azaleastrum* (*R. ovatum*) clusters sister to subgen. *Tsutsusi*, while sect. *Choniastrum* (*R. championiae* and *R. latoucheae*) forms an independent lineage. This finding is corroborated by Shen et al. [15], whose RAD-seq analysis of 144 global species similarly rejected the monophyly of subgen. *Azaleastrum*. However, while their study proposed a weakly supported sister relationship between subgen. *Tsutsusi* and the combined *Azaleastrum*-*Choniastrum* clade, our ddRAD-seq data with substantially higher SNP density and lower missing rates provides decisive evidence for their phylogenetic separation. These results necessitate a taxonomic revision to split subgen. *Azaleastrum*, and reclassify sect. *Azaleastrum* under subgen. *Tsutsusi*.

*R. huadingense* is a deciduous shrub that was jointly described in 1990 by botanist Bingyang Ding and his instructor [17]. It is sparsely distributed in high-altitude mountainous regions and is a key protected wild plant endemic to Zhejiang Province. Due to difficulties in natural regeneration, population decline, and human disturbance leading to a reduced population size, this species was designated as a national Level II key protected wild plant in 2021 [26]. The systematic position of *R. huadingense* has been a topic of long-standing debate due to numerous unique traits. For instance, based on characteristics such as its deciduous nature, soft hairs on young branches, and 3–5 leaf whorls, *R. huadingense* was initially classified in sect. *Brachycalyx* of subgen. *Tsutsusi* when it was first discovered [27]. Cluster analysis using 12 pairs of EST-SSR molecular markers indicated that *R. huadingense* shares the closest phylogenetic relationship with *Rhododendron farrerae*, supporting this classification [28]. However, based on the morphology of flower buds, leaf buds, pollen, and seeds, *R. huadingense* has also been suggested to belong to subgen. *Pentanthera* [29]. This perspective is supported by chloroplast genome data [30], although the results were affected by significantly insufficient sampling within the genus. According to the phylogenomic reconstruction results of this study, *R. huadingense* forms a sister group relationship with subgen. *Tsutsusi*, while it is more distantly related to subgen. *Pentanthera* represented by *R. molle*. Given that the systematic position of *R. huadingense* is now largely clarified at the species level, it is essential to establish this species as a new subgenus.

*R. simsii* var. *putuoense* is a semi-evergreen shrub that is exclusively distributed in the Zhoushan Islands and the eastern coastal areas in Zhejiang. The primary distinguishing feature from the original variety *R. simsii* is the purple corollas of the former, while other morphological characteristics are similar [19]. Consequently, the author previously hypothesized that the phylogenetic relationship between the two should be close. The phylogenetic topology reconstructed in this study supports this hypothesis, indicating that *R. simsii* var. *putuoense* belongs to subgen. *Tsutsusi* and forms a sister group relationship with *R. simsii*; these two taxa have diverged from their most recent common ancestor—*R. simsii*—thereby reinforcing the taxonomic status of the variety.

## 5. Conclusions

This study applied ddRAD-seq to resolve genetic diversity and phylogenetic relationships among *Rhododendron* species in Zhejiang, China. Key findings include: (1) genomic validation of the sister relationship between *R. simsii* and its coastal variety *R. simsii* var. *putuoense* (*F*_st_ = 0.097), highlighting their ecological divergence; (2) recognition of the endemic *R. huadingense* as a distinct subgenus, resolving long-standing taxonomic ambiguities; (3) contrasting genetic diversity patterns, where widespread species (*R. simsii*, π = 0.064–0.068) exhibited higher diversity than endemic/low-altitude species (*R. simiarum*, *R. molle*; π = 0.024–0.033), driven by genetic drift and limited selective pressures; (4) phylogenomic evidence rejecting the monophyly of subgenus *Azaleastrum*, necessitating taxonomic reclassification. The identification of 14 million genome-wide SNPs underscores the efficacy of ddRAD-seq for resolving complex evolutionary histories in non-model plants. These results provide critical insights for biodiversity conservation and emphasize the integration of genomic tools in addressing hybridization and adaptive radiation challenges within *Rhododendron*.

## Figures and Tables

**Figure 1 plants-14-01548-f001:**
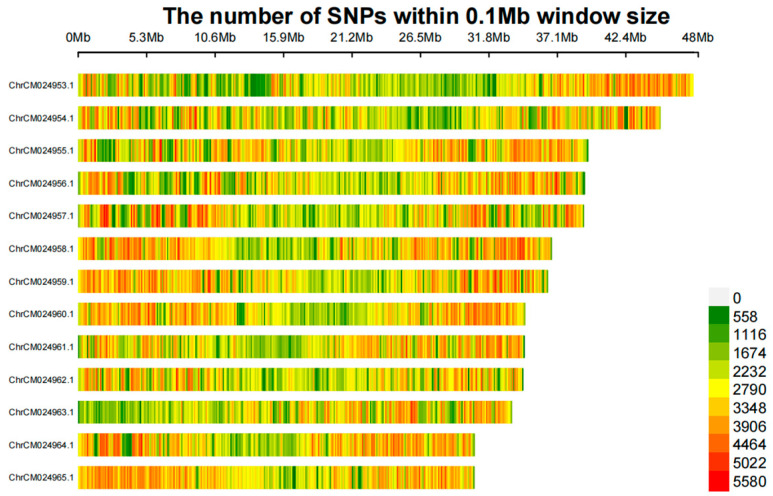
Distribution of SNP loci on thirteen chromosomes of *Rhododendron* genome.

**Figure 2 plants-14-01548-f002:**
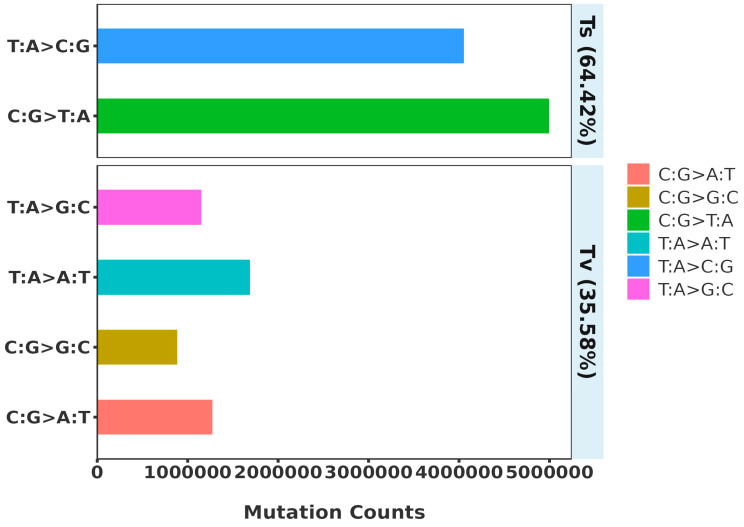
Spectrum of SNP mutation types identified in *Rhododendron* in this study.

**Figure 3 plants-14-01548-f003:**
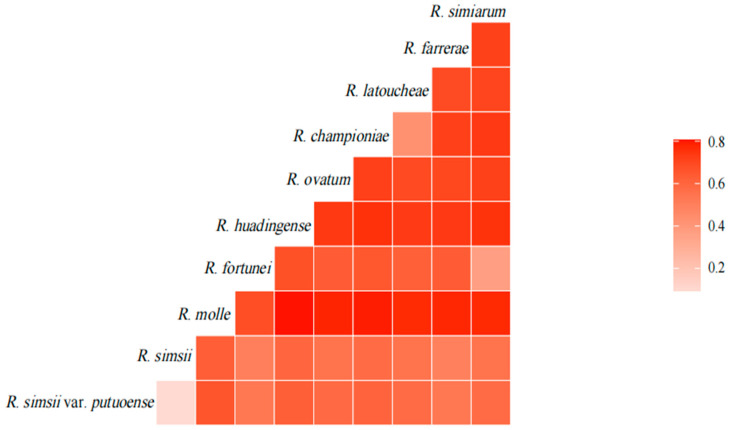
The heat map of pairwise genetic divergence (*F*_st_) between nine species and one variety of *Rhododendron*.

**Figure 4 plants-14-01548-f004:**
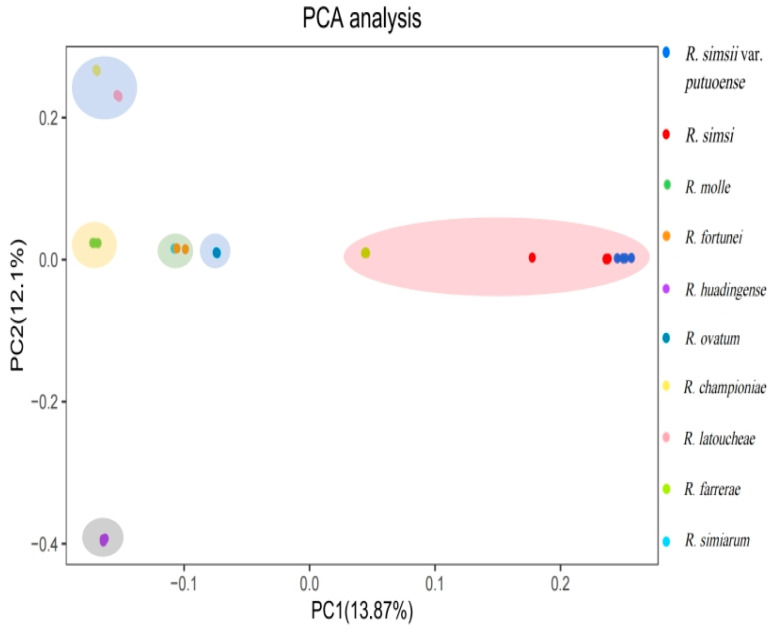
Principle component analysis plot for nine species and one variety of *Rhododendron* species.

**Figure 5 plants-14-01548-f005:**
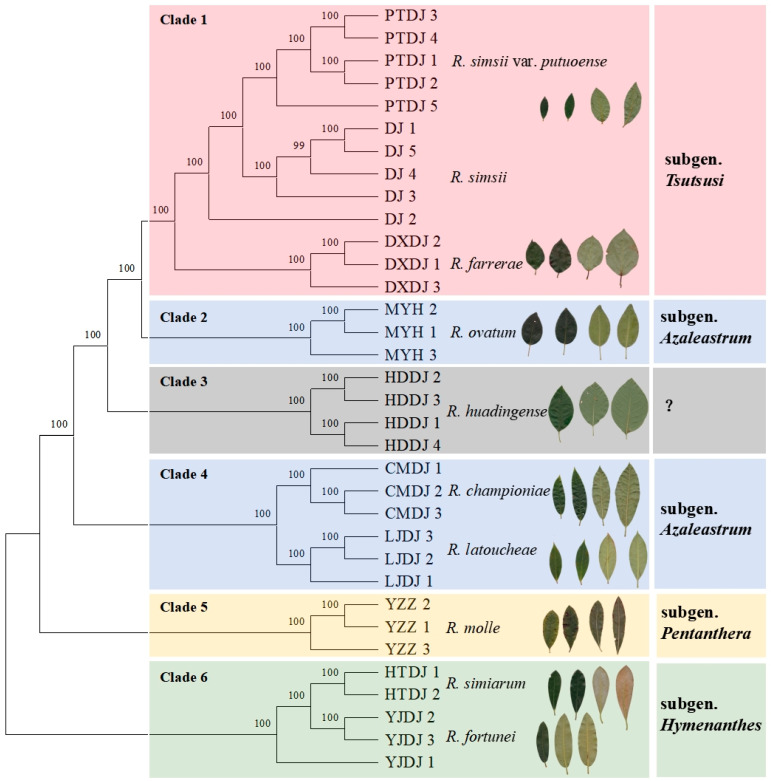
Phylogenomic analysis reconstruction for nine species and one variety from 34 samples of *Rhododendron*.

**Table 1 plants-14-01548-t001:** Data characteristics of nine species and one variety of *Rhododendron* by ddRAD sequencing.

No.	Sample	Species	Subgenus	Section	Reads Number	Total Bases/bp	Mapping Rate/%	Sequence Depth/×
1	DJ_1	*R. simsii*	subgen. *Tsutsusi*	sect. *Tsutsusi*	7,842,910	1,123,802,645	96.84	2.13
2	DJ_2	*R. simsii*	subgen. *Tsutsusi*	sect. *Tsutsusi*	8,447,614	1,209,526,268	96.58	2.29
3	DJ_3	*R. simsii*	subgen. *Tsutsusi*	sect. *Tsutsusi*	6,872,262	983,964,165	96.91	1.86
4	DJ_4	*R. simsii*	subgen. *Tsutsusi*	sect. *Tsutsusi*	8,227,636	1,178,483,283	91.14	2.23
5	DJ_5	*R. simsii*	subgen. *Tsutsusi*	sect. *Tsutsusi*	7,468,622	1,069,413,995	93.78	2.02
6	PTDJ_1	*R. simsii* var. *putuoense*			14,464,694	2,071,406,861	95.01	3.92
7	PTDJ_2	*R. simsii* var. *putuoense*			14,516,774	2,079,276,661	83.70	3.93
8	PTDJ_3	*R. simsii* var. *putuoense*			18,210,666	2,607,244,029	93.94	4.93
9	PTDJ_4	*R. simsii* var. *putuoense*			9,344,540	2,106,199,392	94.93	3.98
10	PTDJ_5	*R. simsii* var. *putuoense*			7,083,544	1,337,742,896	94.50	2.53
11	DXDJ_1	*R. farrerae*	subgen. *Tsutsusi*	sect. *Brachycalyx*	7,887,388	1,129,332,945	95.82	2.14
12	DXDJ_2	*R. farrerae*	subgen. *Tsutsusi*	sect. *Brachycalyx*	7,065,322	1,011,327,506	94.25	1.91
13	DXDJ_3	*R. farrerae*	subgen. *Tsutsusi*	sect. *Brachycalyx*	7,406,124	1,060,945,460	95.96	2.01
14	HDDJ_1	*R. huadingense*	subgen. *Tsutsusi*	sect. *Brachycalyx*	7,345,694	1,051,336,075	92.74	1.99
15	HDDJ_2	*R. huadingense*	subgen. *Tsutsusi*	sect. *Brachycalyx*	7,588,046	1,085,558,978	93.62	2.05
16	HDDJ_3	*R. huadingense*	subgen. *Tsutsusi*	sect. *Brachycalyx*	8,763,824	1,254,026,187	94.01	2.37
17	HDDJ_4	*R. huadingense*	subgen. *Tsutsusi*	sect. *Brachycalyx*	7,032,948	1,006,765,697	93.28	1.90
18	YZZ_1	*R. molle*	subgen. *Pentanthera*	sect. *Pentanthera*	6,872,262	954,249,606	89.29	1.81
19	YZZ_2	*R. molle*	subgen. *Pentanthera*	sect. *Pentanthera*	8,227,636	1,071,341,272	88.80	2.03
20	YZZ_3	*R. molle*	subgen. *Pentanthera*	sect. *Pentanthera*	7,468,622	935,457,901	90.03	1.77
21	MYH_1	*R. ovatum*	subgen. *Azaleastrum*	sect. *Azaleastrum*	7,156,964	1,024,609,279	95.74	1.94
22	MYH_2	*R. ovatum*	subgen. *Azaleastrum*	sect. *Azaleastrum*	7,987,170	1,143,354,013	94.15	2.16
23	MYH_3	*R. ovatum*	subgen. *Azaleastrum*	sect. *Azaleastrum*	6,943,508	994,044,215	94.69	1.88
24	CMDJ_1	*R. championiae*	subg. *Azaleastrum*	sect. *Choniastrum*	7,001,716	1,002,884,795	91.92	1.90
25	CMDJ_2	*R. championiae*	subgen. *Azaleastrum*	sect. *Choniastrum*	7,083,544	1,014,407,528	91.71	1.92
26	CMDJ_3	*R. championiae*	subgen. *Azaleastrum*	sect. *Choniastrum*	6,998,250	1,002,294,502	90.80	1.90
27	LJDJ_1	*R. latoucheae*	subgen. *Azaleastrum*	sect. *Choniastrum*	7,988,094	1,144,587,841	92.62	2.17
28	LJDJ_2	*R. latoucheae*	subgen. *Azaleastrum*	sect. *Choniastrum*	6,100,640	873,849,781	92.48	1.65
29	LJDJ_3	*R. latoucheae*	subgen. *Azaleastrum*	sect. *Choniastrum*	8,249,558	1,181,825,031	91.94	2.24
30	HTDJ_1	*R. simiarum*	subgen. *Hymenanthes*	subt. *Argyrophylla*	7,336,492	1,050,798,076	90.36	1.99
31	HTDJ_2	*R. simiarum*	subgen. *Hymenanthes*	subt. *Argyrophylla*	8,418,280	1,205,949,808	88.20	2.28
32	YJDJ_1	*R. fortunei*	subgen. *Hymenanthes*	subt. *Fortunea*	6,998,250	1,071,807,669	90.95	2.03
33	YJDJ_2	*R. fortunei*	subgen. *Hymenanthes*	subt. *Fortunea*	7,842,910	980,284,260	92.32	1.85
34	YJDJ_3	*R. fortunei*	subgen. *Hymenanthes*	subt. *Fortunea*	8,447,614	997,989,300	91.97	1.89

The infrageneric taxon system of *Rhododendron* follows classification outlined in the Flora of China (2005).

**Table 2 plants-14-01548-t002:** Genetic diversity statistics for nine species and one variety of *Rhododendron* species.

Species	Num Indv	*H* _o_	*H* _e_	π	*F* _is_
*R. simsii* var. *putuoense*	4.8102	0.0549	0.0574	0.0642	0.0199
*R. simsii*	4.7006	0.0474	0.0611	0.0685	0.0466
*R. molle*	2.8409	0.0293	0.0199	0.0241	−0.0087
*R. fortunei*	2.7522	0.0362	0.0432	0.0528	0.0306
*R. huadingense*	3.7156	0.0345	0.0280	0.0323	−0.0035
*R. ovatum*	2.8092	0.0355	0.0301	0.0365	0.0019
*R. championiae*	2.8421	0.0379	0.0301	0.0364	−0.0023
*R. latoucheae*	2.8224	0.0372	0.0318	0.0386	0.0028
*R. farrerae*	2.7556	0.0318	0.0325	0.0395	0.0140
*R. simiarum*	1.9170	0.0374	0.0245	0.0330	−0.0066
Mean value	3.1966	0.0382	0.0359	0.0426	0.0193

## Data Availability

The raw sequencing data of *Rhododendron* has been deposited to Sequence Read Archive (SRA) database in NCBI with BioProject ID PRJNA1231201.

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
