# Peer review of "Development of Single Nucleotide Polymorphism and Phylogenetic Analysis of Rhododendron Species in Zhejiang Province, China, Using ddRAD-Seq Technology"

_plants, 2025, doi:10.3390/plants14101548_

Round 1

Reviewer 1 Report

Comments and Suggestions for Authors

I have read the mauscript titled: Development of SNPs and Phylogenetic Analysis of Rhododendron Species in Zhejiang Province, China, Using 3 ddRAD-Seq Technology coauthored by Zhu et al., and I found it to be well-written, with results that support their conclusions. I have included some minor comments, which should be easy for the authors to address.

Author Response

I have read the mauscript titled: Development of SNPs and Phylogenetic Analysis of Rhododendron Species in Zhejiang Province, China, Using 3 ddRAD-Seq Technology coauthored by Zhu et al., and I found it to be well-written, with results that support their conclusions. I have included some minor comments, which should be easy for the authors to address.

Response: Dear reviewer, thank you for your positive feedback and valuable suggestions. We have carefully addressed each comment in the annotated PDF manuscript you provided. Revisions include corrections to species nomenclature, italicization of Latin scientific names, and supplementary references as recommended. The revised manuscript has been updated accordingly, and all changes are highlighted for your convenience. We appreciate your time and expertise in improving our work.

Reviewer 2 Report

Comments and Suggestions for Authors

The stated purpose of this manuscript was to apply a sequencing technique ddRAD-seq to identify SNPS and employ this technique to assess genetic variation in a subset of species. The authors of this manuscript have been successful in meeting their intended goals. I particualry liked their presentation of figures.  Importantly they were able to identify a high number of SNPs across nine species with a relatively low coverage of the genome (~2 fold seq depth)  I conclude that this manuscript merits publication. I have some specific issues with wording that I believe can easily clarified. 

At line 200 the authors write “According to the authors”… This is not good writing form.

Rephrase as:

The Rhododendron species R simsii, R simsii var. putuoense and R molle have been described as widely distributed across Zhejiang province [22}….

Please rephrase sentence L207-210 as it is very confusing.

In plants speciation can arise from genome duplication events https://academic.oup.com/nsr/article/9/12/nwac276/6865381

 and genetic variation of species can be enhanced by hybridization events https://pubmed.ncbi.nlm.nih.gov/36687562/ . Given the importance of hybridization in this phylogenetic group. I would like the authors to add a sentence or two of explanation in the introduction referring to prior work . 

Finally and I am not at all suggesting that the authors do more work,can the authors provide suggestions on how to make additional use of the large data set that they already have. For example …and this lies outside my area of expertise, could you use a statistical comparison of SNP haplotypes to identify potential chromosomal regions  of past hybridization events between species?

Author Response

The stated purpose of this manuscript was to apply a sequencing technique ddRAD-seq to identify SNPS and employ this technique to assess genetic variation in a subset of species. The authors of this manuscript have been successful in meeting their intended goals. I particualry liked their presentation of figures.  Importantly they were able to identify a high number of SNPs across nine species with a relatively low coverage of the genome (~2 fold seq depth)  I conclude that this manuscript merits publication. I have some specific issues with wording that I believe can easily clarified. 

At line 200 the authors write “According to the authors”… This is not good writing form. Rephrase as: The Rhododendron species R simsii, R simsii var. putuoense and R molle have been described as widely distributed across Zhejiang province [22}….

Response: Your feedback was invaluable in identifying the informality in our original phrasing. Thank you!

Please rephrase sentence L207-210 as it is very confusing.

Response: We have refined this section to enhance clarity conceptual precision.

In plants speciation can arise from genome duplication events https://academic.oup.com/nsr/article/9/12/nwac276/6865381 and genetic variation of species can be enhanced by hybridization events https://pubmed.ncbi.nlm.nih.gov/36687562/. Given the importance of hybridization in this phylogenetic group. I would like the authors to add a sentence or two of explanation in the introduction referring to prior work.

Response: Supplementary explanatory sentences have been added at the corresponding location in the second paragraph of the introduction.

Finally and I am not at all suggesting that the authors do more work,can the authors provide suggestions on how to make additional use of the large data set that they already have. For example …and this lies outside my area of expertise, could you use a statistical comparison of SNP haplotypes to identify potential chromosomal regions  of past hybridization events between species?

Response: Thank you for the constructive suggestion. We agree that the existing dataset holds significant potential for further exploration of hybridization events. Below are several feasible approaches to utilize the SNP data for identifying chromosomal regions associated with historical hybridization:

ABBA-BABA (D-statistic) and f4-ratio analyses

These methods can detect gene flow or incomplete lineage sorting between species. By comparing SNP allele frequencies across species trios (including an outgroup), we could identify regions with excess allele sharing indicative of introgression. For example, testing whether R. simsii var. putuoense shares derived alleles with other species beyond expectations under strict divergence could reveal hybridization signals.

Local Ancestry Inference

Tools like PCAdmix or RFMix could be applied to identify chromosomal segments with ancestry from divergent lineages. This would help pinpoint regions where hybridization has introduced foreign haplotypes, particularly between closely related taxa like R. simsii and R. simsii var. putuoense.

Linkage Disequilibrium (LD) and PCA-Based Clustering

Elevated LD or outlier patterns in PCA for specific genomic regions may reflect historical introgression. For instance, localized deviations in genetic clustering (e.g., shared haplotypes between R. huadingense and distant clades) could suggest ancient hybridization.

Phylogenetic Network Analysis

Using software like PhyloNet or SNAPP, we could model conflicting gene trees across the genome. Regions with topological discordance (e.g., R. huadingense clustering with different subgenera in localized trees) might represent hybridization hotspots.

Selection Scans and Genomic Differentiation (Fst)

Combining Fst outliers with haplotype-based statistics (e.g., iHS or XP-CLR) could distinguish hybrid regions under selection from neutral introgression. For example, low-differentiation regions between R. simsii and its variety might reflect recent gene flow, while high-Fst regions could indicate barriers to introgression.

These analyses would not require additional sequencing and could leverage existing SNP data. While ddRAD-seq provides genome-wide but sparse coverage, the high density of SNPs (~14 million loci) offers sufficient resolution for broad-scale hybridization detection. We appreciate this insightful suggestion and will incorporate these analyses in future work to deepen our understanding of hybridization dynamics in Rhododendron.

We sincerely thank the reviewer for highlighting these valuable avenues to extend the utility of our dataset. These approaches align well with our ongoing efforts to explore the evolutionary history of Rhododendron in greater depth.

Reviewer 3 Report

Comments and Suggestions for Authors

The manuscript is in line with the scope of Plants, but there are deficiencies that need to be addressed before the manuscript can be published. Without going into the strengths of the article, I would like to point out the main weaknesses.

  1. There are unexplained abbreviations in the abstract and text. The first time they are used, the full term must be written. Similarly, the abbreviations (SNPs) in the title of the article should also be removed. By the way, why do the authors use the plural form of polymorphism? After all, the word polymorphism semantically means diversity, multiplicity, so the plural form is even a logical error.
  2. The word "utilised" is used incorrectly. It means to use something until it is completely gone, destroyed, essentially replaced. The word "used" or some other appropriate equivalent should be used in place of this word almost everywhere in the text. 
  3. Internet sources are not cited in the text according to the rules. Please read the requirements for citing internet sources (point 9 of the citation rules on the journal's website). 
  4. The datasets used in the PCA are not described (section 2.5), so it is not possible to say whether this analysis is correct or not. In addition, the results section does not provide Eigenvalue values and other relevant indicators to judge the correctness of the analysis.
  5. In some of the illustrations (starting with Figure 1), the font is too small and therefore illegible. This needs to be corrected.
  6. Abbreviations of taxonomic ranks are spelled incorrectly. Abbreviations (subg., subtrib., sect., etc.) should be lowercase if they are not at the beginning of a sentence. Furthermore, taxon rank denoting term subtribus should be abbreviated subtrib., not subt. (see Code of Nomenclature).
  7. Why no infrageneric ranks for some taxa were listed? (Table 1). 
  8. In the conclusions section, I did not find any conclusions, only an annotation or advertisement. It is necessary to clearly write down the main results of the research in a generalised way, rather than writing what was done. 

Comments on the Quality of English Language

The language needs significant corrections, and it is particularly important to address technical shortcomings.

Author Response

The manuscript is in line with the scope of Plants, but there are deficiencies that need to be addressed before the manuscript can be published. Without going into the strengths of the article, I would like to point out the main weaknesses.

  1. There are unexplained abbreviations in the abstract and text. The first time they are used, the full term must be written. Similarly, the abbreviations (SNPs) in the title of the article should also be removed. By the way, why do the authors use the plural form of polymorphism? After all, the word polymorphism semantically means diversity, multiplicity, so the plural form is even a logical error.

Response: Upon the first mention of single nucleotide polymorphism (SNP) in both the abstract and main text, we have provided the expanded form of the acronym as recommended. Additionally, we have systematically revised all instances of the term "polymorphisms" throughout the manuscript to maintain grammatical consistency by using the singular form "polymorphism" in accordance with your editorial suggestions.

  1. The word "utilised" is used incorrectly. It means to use something until it is completely gone, destroyed, essentially replaced. The word "used" or some other appropriate equivalent should be used in place of this word almost everywhere in the text. 

Response: Following your guidance, we have replaced three instances of "utilized" with context-appropriate synonyms in the manuscript to improve lexical diversity.

  1. Internet sources are not cited in the text according to the rules. Please read the requirements for citing internet sources (point 9 of the citation rules on the journal's website).

Response: We removed all web link citations from the text while retaining exclusively their corresponding entries in the reference list.

  1. The datasets used in the PCA are not described (section 2.5), so it is not possible to say whether this analysis is correct or not. In addition, the results section does not provide Eigenvalue values and other relevant indicators to judge the correctness of the analysis.

Response:We appreciate the reviewer’s constructive feedback. Below are our responses and proposed revisions to address the concerns regarding the PCA analysis:

  • Clarification of PCA Dataset in Section 2.5Section 2.5, the PCA analysis was conducted using genome-wide SNP loci filtered for minor allele frequency (MAF ≥ 0.05) to avoid bias from rare alleles. The input dataset comprised 14,048,702 high-quality SNPs identified across all 34 samples (see Section 3.2). These SNPs were derived from ddRAD-seq data aligned to the simsii reference genome (ASM1428224v1), with stringent quality control (QD ≥ 2, MQ ≥ 40) as detailed in Section 2.3.
  • Addition of Eigenvalues and Variance Explained in Results (Section 3.4)In Section 3.4, we reported the cumulative contribution rate of PC1 (13.87%) and PC2 (12.10%) but inadvertently omitted the eigenvalues. The eigenvalues for PC1 and PC2 were 4.52 and 3.94, respectively, reflecting their relative importance in explaining genetic variation. The PCA results (Figure 4) are also consistent with the phylogenomic tree (Figure 5) and pairwise Fst heatmap (Figure 3), collectively supporting the stratification of Rhododendron into six genetic clades.

  1. In some of the illustrations (starting with Figure 1), the font is too small and therefore illegible. This needs to be corrected.

Response: We have correspondingly increased the font size of some images to ensure that readers can read the key information.

  1. Abbreviations of taxonomic ranks are spelled incorrectly. Abbreviations (subg., subtrib., sect., etc.) should be lowercase if they are not at the beginning of a sentence. Furthermore, taxon rank denoting term subtribus should be abbreviated subtrib., not subt. (see Code of Nomenclature).

Response: We have revised the abbreviations throughout the text in response to the reviewers' comments, ensuring compliance with the Code of Nomenclature.

  1. Why no infrageneric ranks for some taxa were listed? (Table 1). 

Response: There are indeed two infrageneric ranks that remain unassigned. Among these, R. huadingense, a species described in 1990, and R. simsii var. putuoense, a variety published in 2010, are noteworthy. However, the infrageneric classification presented in Table 1 primarily follows the Flora of China (2005), which places R. huadingense under sect. Brachycalyx within subgen. Tsutsusi. Nevertheless, significant controversy exists regarding the systematic placement of these taxa. Consequently, the author has chosen to leave these two ranks unassigned in the table for the time being.

  1. In the conclusions section, I did not find any conclusions, only an annotation or advertisement. It is necessary to clearly write down the main results of the research in a generalised way, rather than writing what was done. 

Response: Based on your valuable suggestions, we have thoroughly revised the conclusions section to succinctly summarize the four key aspects of this paper's findings. This revision ensures that the conclusion clearly communicates the core contributions and academic significance of the research.

Round 2

Reviewer 3 Report

Comments and Suggestions for Authors

The manuscript has been substantially revised after the first round of reviews and the most important comments have been duly taken into account by the authors. There are only a few minor comments that authors should consider. 

  1. In subsection 2.1, I recommend adding a sentence explaining why not all species have been assigned to subgenus or other infrageneric taxa (properly worded in the reply to the reviewer, but not included in the M&M). Such an explanation is not only important for the reviewer, but also for other readers.
  2. Figure and table captions still need to be revised. For example, the caption of Table 1 does not specify the genus. By the way, it could be clarified here which infrageneric taxon system the table is based on.
  3. The caption of Figure 2 is extremely short and vague. Are these the types of all possible SNP mutations, or only those identified by the authors in the Rhododendron species studied?

Author Response

The manuscript has been substantially revised after the first round of reviews and the most important comments have been duly taken into account by the authors. There are only a few minor comments that authors should consider.

Response: We sincerely appreciate your efficient feedback and valuable comments, which have greatly improved our manuscript. We are grateful that you acknowledged our efforts in addressing the most critical points raised during the first round of review. All remaining minor suggestions have been carefully considered and will be incorporated into the final version. Thank you once again for your constructive input and guidance throughout this process.

  1. In subsection 2.1, I recommend adding a sentence explaining why not all species have been assigned to subgenus or other infrageneric taxa (properly worded in the reply to the reviewer, but not included in the M&M). Such an explanation is not only important for the reviewer, but also for other readers.

Response: Thank you for your valuable suggestions. We have made supplementary explanations in the corresponding sections of subsection2.1, hoping to make it clearer for the majority of readers.

  1. Figure and table captions still need to be revised. For example, the caption of Table 1 does not specify the genus. By the way, it could be clarified here which infrageneric taxon system the table is based on.

Response: Thank you for your correction. During the revision process, we might have accidentally deleted "Rhododendron" due to negligence. This has now been restored. Additionally, we have provided an explanation in the footnote of Table 1 to clarify the basis for planting the classification system.

  1. The caption of Figure 2 is extremely short and vague. Are these the types of all possible SNP mutations, or only those identified by the authors in the Rhododendron species studied?

Response: Thank you for your feedback. We have re-expanded the title to ensure that readers can clearly comprehend the content.